# Initial agronomic benefits of enhanced weathering using basalt: A study of spring oat in a temperate climate

**Kirstine Skov**[1]*, **Jez Wardman**[1], **Matthew Healey**[1], **Amy McBride**[1], **Tzara Bierowiec**[1], **Julia Cooper**[2], **Ifeoma Edeh**[1], **Dave George**[3], **Mike E. Kelland**[4], **Jim Mann**[1], **David Manning**[3], **Melissa J. Murphy**[1], **Ryan Pape**[1], **Yit A. Teh**[3], **Will Turner**[1], **Peter Wade**[1], **Xinran Liu**[1]

1 UNDO Carbon Ltd, London, United Kingdom, 2 Organic Research Centre, Cirencester, Gloucestershire, United Kingdom, 3 School of Natural and Environmental Sciences, Newcastle University, Newcastle upon Tyne, Tyne and Wear, United Kingdom, 4 School of Biosciences, The University of Sheffield, Sheffield, United Kingdom

* kirstine.skov@un-do.com

**Data Availability Statement:** The data is shared in the public repository Zenodo: 10.5281/zenodo. 8158681.

## Abstract

Addressing soil nutrient degradation and global warming requires novel solutions. Enhanced weathering using crushed basalt rock is a promising dual-action strategy that can enhance soil health and sequester carbon dioxide. This study examines the short-term effects of basalt amendment on spring oat (*Avena sativa L*.) during the 2022 growing season in NE England. The experimental design consisted of four blocks with control and basalt-amended plots, and two cultivation types within each treatment, laid out in a split plot design. Basalt (18.86 tonnes ha$^{-1}$) was incorporated into the soil during seeding. Tissue, grain and soil samples were collected for yield, nutrient, and pH analysis. Basalt amendment led to significantly higher yields, averaging 20.5% and 9.3% increases in direct drill and ploughed plots, respectively. Soil pH was significantly higher 256 days after rock application across cultivation types (direct drill: on average 6.47 vs. 6.76 and ploughed: on average 6.69 vs. 6.89, for control and basalt-amended plots, respectively), likely due to rapidly dissolving minerals in the applied basalt, such as calcite. Indications of growing season differences in soil pH are observed through direct measurement of lower manganese and iron uptake in plants grown on basalt-amended soil. Higher grain and tissue potassium, and tissue calcium uptake were observed in basalt-treated crops. Notably, no accumulation of potentially toxic elements (arsenic, cadmium, chromium, nickel) was detected in the grain, indicating that crops grown using this basaltic feedstock are safe for consumption. This study indicates that basalt amendments can improve agronomic performance in sandy clay-loam agricultural soil under temperate climate conditions. These findings offer valuable insights for producers in temperate regions who are considering using such amendments, demonstrating the potential for improved crop yields and environmental benefits while ensuring crop safety.

**Funding:** The study design and yield data collection was conducted by Newcastle University, independent of funders of the project. The funders (UNDO Carbon Ltd) were involved in all other data collection (including soil sampling and plant tissue sampling) and analysis, conducted in collaboration with Newcastle University. Decision to publish and preparation of the manuscript was done in collaboration between Newcastle University and UNDO Carbon Ltd. UNDO Carbon Ltd. provided support in the form of salaries for Kirstine Skov, Jez Wardman, Matthew Healey, Amy McBride, Tzara Bierowiec, Ifeoma Edeh, Melissa J. Murphy, Ryan Pape, Will Turner, Peter Wade, Mike Kelland, Jim Mann and Xinran Liu. The specific roles of these authors are articulated in the 'author contributions' section.

**Competing interests:** The authors have read the journal's policy and have the following competing interests: Kirstine Skov, Jez Wardman, Matthew Healey, Amy McBride, Tzara Bierowiec, Ifeoma Edeh, Melissa J. Murphy, Ryan Pape, Will Turner, Peter Wade and Xinran Liu are presently or were recently employed in the UNDO Carbon Ltd Science and Research department and may hold options in UNDO Carbon Ltd. Mike Kelland is an independent consultant for UNDO Carbon Ltd. Jim Mann is the founder and CEO of UNDO Carbon Ltd. David Manning is part of UNDO Carbon Ltd.s scientific advisory board. Julia Cooper, Dave George and Yit A. Teh declare no conflicts of interest. This does not alter our adherence to PLOS ONE policies on sharing data and materials.

## Introduction

Nutrient deficiencies in agricultural soil represent a global challenge and are of major concern in relation to sustaining and increasing crop yields, thereby ensuring food security to feed the growing human population [1]. Furthermore, the availability of new agricultural land is limited [2] and the effects of climate change are predicted to impede food production [3]. Enhanced weathering (EW) of silicate rock may aid in securing food safety, through agronomic benefits resulting from the release of rock-derived nutrients [4, 5]. EW has also been proposed as a promising, scalable carbon dioxide removal (CDR) technology with the global potential to sequester gigatons of atmospheric carbon dioxide [5, 6].

The use of crushed silicate rocks, such as basalt, as a soil amendment on croplands is associated with a range of agronomic benefits [7–9]. Studies dating back to the 1960's [10, 11] and 1930's [12, 13] reveal the potential for yield enhancement and soil improvement following the application of crushed basalt rock to agricultural and forest soils. However, some studies utilising volcanic rock dust have shown little or no agronomic benefit [14]. In this particular trial [14], the rock type used was not described, and although it appears not to be basaltic on the basis of its high iron content, this study may have been influential in deterring interest in the agricultural use of crushed basalt. However, clear benefits for plant growth have been shown in more rigorous experiments [15], in which the crushed rock was well defined as basaltic on the basis of its chemical and mineralogical composition. Mafic rocks, such as basalt, are defined mineralogically as containing pyroxene and plagioclase-group minerals [16] and are defined chemically as containing <52 wt.% $SiO_2$ and <5 wt.% alkali metal oxides ($Na_2O$ and $K_2O$) [17]. By these definitions, the composition of mafic rocks can exhibit substantial variation across different geographical locations and geological ages [16, 18]. This variability makes it extremely important to characterise the mineralogy and chemistry of each rock used within experiments.

Until very recently, interest in the agronomic benefits of basalt has been limited; the use of this rock type in EW studies has provided a new and urgent incentive for rigorous trials that investigate the effects on crop growth. Basalt can be rich in minerals that release macro- (e.g. calcium, magnesium, silicon, potassium, phosphorus, sulfur, but excluding nitrogen) and micro-nutrients (e.g. copper, iron, manganese, molybdenum and zinc) during dissolution that are essential for plant growth [4–6]. Crushed silicate rocks have considerable potential to ameliorate potassium and micronutrient deficiencies, following agricultural soil nutrient stripping [9]. Some of these micronutrients are redox-active, which makes them important cofactors in enzymatic, metabolic and cellular processes [19]. Additionally, these micronutrients are actively involved in nitrogen fixation and photosynthesis [20, 21].

Increased soil nutrient availability may lead to increased crop yields. A synthesis study [9] found a significant yield increase in 19 out of 34 peer-reviewed papers. However, the majority (i.e. 14) of these studies were conducted on highly weathered, acidic soils in tropical regions. Comparatively, studies in temperate soils showed less definitive evidence of yield gains, with the exception of experiments where faster weathering minerals (e.g. nepheline and biotite mica) were applied. Improvements in maize and potato yields have also been reported recently in a study using glacial rock flour in the temperate climate of southern Denmark [22].

While macro- and micro-nutrients are present within silicate rocks, they are not immediately available for plant uptake; they only become available following mineral dissolution. The slower nutrient release rate may, however, be a positive trait of silicate rock amendments compared to conventional (synthetic) fertilisers, as this may reduce the likelihood of nutrient leaching or surface run-off which can subsequently contaminate groundwater and water courses [8]. Simulations using a reactive transport model and the mineralogical composition

of six commercially available basaltic quarry fines suggest that some basalts have the potential to release sufficient phosphorus to substitute average P-fertiliser application to tillage crops in the UK and the United States after one year [18].

The most pronounced potential agronomic benefit of crushed silicate rock as a soil amendment is probably the neutralising effect on pH in acidic soils, as a result of alkaline products generated from mineral dissolution [4]. Increased pH in acidic soils generally increases plant nutrient availability by reducing the formation of insoluble nutrient compounds caused by soil acidification [4], hence potentially increasing crop yield. Previous studies have found pH increases ranging between a relatively modest 0.2 pH units up to as much as 2 pH units on soils where the starting pH was around 4.5 [9, 23, 24] following the application of crushed rock.

While silicate rocks contain many essential elements for crop nutrition, the release and accumulation of potentially toxic trace elements (PTEs) is an important consideration where EW amendments are applied to agricultural soils [25]. Some PTEs are beneficial plant micro-nutrients in low concentrations, but become potentially harmful at elevated concentrations [25]. Soil accumulation of nickel and chromium are of particular concern due to the potential risks of harmful effects on the environment and human health [25]. Nickel and chromium concentrations in ultramafic minerals like olivine (approximately 1300 and 2300 mg kg$^{-1}$, respectively, [26]) have raised particular concerns in relation to application on agricultural land [27, 28]. Although levels are an order of magnitude lower in basalt [29], nickel concentrations may exceed the EU soil improver limit (100 mg kg$^{-1}$ [30]), especially if the rock is sourced from the North-Atlantic igneous province [25]. Silicate rocks may also include trace amounts of other potentially hazardous elements, such as arsenic, cadmium, mercury and lead [31].

Soil health is a vital part of effective farming and has drawn a greater focus in recent years due to decades of intensive farming practices that have led to soil degradation [32]. An increasing number of farmers are changing field management practices from ploughing to no-till/direct drilling, in order to reduce soil erosion and maintain natural soil fertility. Both cultivation types have implications for the physical and biological conditions of the soil, and may therefore influence the rate of basalt dissolution. Direct drilling forms the basis of no-till cultivation approaches, which involve planting the seeds directly into the soil with reduced disturbance to the soil surface [33]. Ploughing, on the other hand, is a more invasive form of tillage, which involves turning over the ground in order to loosen the soil, break up clods, and incorporate organic matter. Compared to ploughing, direct drilling may improve the soil health and thereby yield, by markedly increasing the abundance of earthworms and ability of the soil to retain organic matter and soil nutrients [33, 34]. However, depending on the soil type and condition, both direct drilling and ploughing may lead to compaction of the topsoil with reduced root penetration depth [35] and loss in yield as a consequence [36, 37]. In this study the effect of basalt amendment is evaluated across both direct drill and ploughed plots.

As a land-based CDR technology, the application of EW amendments to agricultural land is projected to be highly scalable due to the abundance [38] and availability of basalt [39]. However, successful large-scale deployment of EW depends on landowners' willingness to grant access to their land or apply basalt themselves. The value proposition of EW as a CDR technology for landowners is likely to improve when its safe practice and positive agronomic benefits, especially in terms of crop yield and soil pH, are demonstrated under climate, soil, crop, and cultivation conditions similar to those experienced by the farmers themselves. This study aimed to investigate the agronomic effects of basalt amendment on crop yield (spring oat, *Avena sativa* L.), nutrient and PTE uptake as well as soil pH, during the first growing season in a temperate climate. The trial was carried out across two common cultivation types in an agriculturally intensive area of NE England, during the 2022 growing season.

## Methods

### Site description

The study was undertaken at the Newcastle University-managed Nafferton Farm, located in north-east England, UK (54˚59'07."N, 1˚53'59.4"W). All sampling permits were obtained through Newcastle University. The soil is predominantly classified as a uniform Dystric Stagnosol [40], with a sandy clay loam soil texture (Sand: 60.5%, Silt: 22.5% and Clay: 17.0%) [41]. The soil is slightly acidic (average pH in 2011 of 6.3) [41], with an average soil organic carbon stock of 60.64 ±0.9 MgC ha$^{-1}$ [40]. The field has a slight gradient in the west to east direction, with a slope corresponding to 2.8% parallel to the plots. The site is located in the Köppen climate classification subtype 'marine west coast climate' [40]. Weather station data (from 2002 to 2022) was recorded on site, using an automatic weather station (S1 Fig).

### Experimental design

The trial was conducted on the farm's Quality Low Input Food (QLIF) plots. The plots were originally established in 2001 and have historically been used to investigate differences in organic and conventional farming (e.g. [41]). The experimental design consists of four blocks, each containing a single replicate of four different treatments, of which two were relevant to this study; the control and basalt-amended plots (Fig 1). Within the context of the QLIF platform, the control plots are managed according to standard commercial practice. Nested within each treatment are two different cultivation types; no-till/direct drilling and ploughing, see Fig 1.

### Field management

During the 2022 growing season, spring oat [42], was cultivated on the plots. The ploughed fields were tilled to a depth of 15–30 cm at the end of March, followed by a press within three days (using Simba Unipress). Basalt was applied on 12th April 2022 using a conventional agricultural lime spreader. Although the target application rate was set to 20 tonnes ha$^{-1}$, as this is

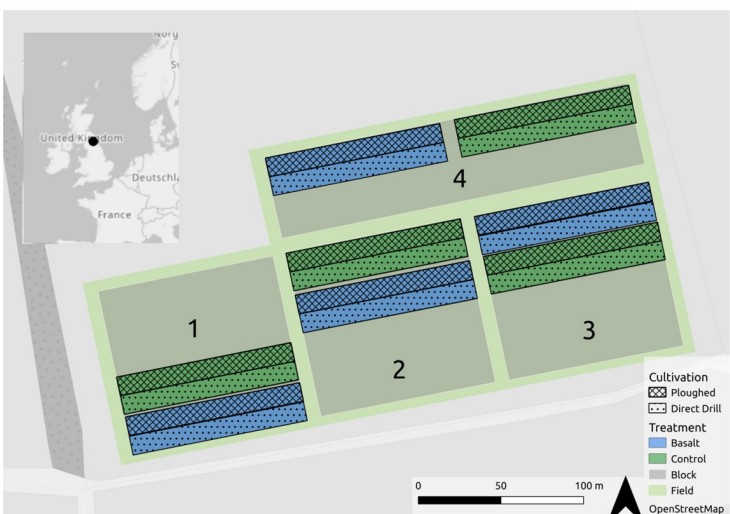

**Fig 1. Experimental plot design.** The site includes four blocks (transparent grey), each 96 x 96 meters. Within each block there are two treatments: Control (green) and basalt-amended plots (blue). For each treatment there are two cultivation types: no-till/direct drill (dotted) and ploughed (hashed). Each cultivation plot is 12 x 96 meters. Background: OpenStreetMap. Inset: location of the Nafferton Research Farm in NE England, overlaid OpenStreetMap.

the maximum delivery the commercial spreader could achieve in a single pass, the actual mean spreading rate was determined to be 18.86 tonnes ha$^{-1}$. Ploughing was carried out prior to basalt application, hence, the basalt was exclusively incorporated to the seeding depth across both cultivation types. No further efforts were made to incorporate the basalt into the soil, as commercial methods were employed to ensure the process accurately represented farm-scale operations. Both control and basalt-amended plots were fertilised with liquid NitroFlow Urea Ammonium Nitrate (UAN) on 27th April 2022, with an application rate of 246 L ha$^{-1}$, providing 75.5 kgN ha$^{-1}$ and 23.6 kgS ha$^{-1}$. The plots were additionally treated with conventional applications of herbicides and fungicides (S1 Appendix).

The seeding rate across both cultivation types was 180 kg ha$^{-1}$, where seeds were drilled into the soil using a John Deere 750A seed drill at a target depth of 2 cm on 15th April 2022. For both the direct drill and ploughed plots, the seed beds were rolled within two days (using a Cambridge roller). All fields were harvested on 16th September 2022 using a plot combine (Deutz-Fahr M660).

## Soil sampling and analysis

Soil samples were collected on 1st/2nd February 2023, prior to any management activities for the impending growing season. Historically, each cultivation plot had been divided into several different treatments, with four evenly-spaced subplots [43]. These subplots were abandoned in 2017, but in order to avoid any potential legacy bias in soil pH, four soil samples were collected from each subplot for the current study, resulting in a total number of 16 soil samples from each cultivation plot, 64 samples from each block and 256 samples in total. The samples were taken to a depth of 15 cm and dried for five days at 40˚C. The samples were then sieved to obtain the <2 mm fraction and analysed for pH using a 1:2.5 mixing ratio of soil to H$_2$O. The pH was determined using a Metrohm 914 pH/conductometer.

## Plant tissue and grain sampling

The plant tissue sampling was conducted on 1st August 2022, by collecting 100 fully expanded flag leaves from the centerline of each of the 16 plots. The crops were at growth stage 77–81 at the time of sampling, just prior to the onset of senescence. The fresh samples were packaged in plastic zip lock bags, stored at ambient temperature and dispatched for laboratory analysis on the same day. The samples were received at the laboratory on 3rd August 2022. The tissue samples were dried at 60˚C and milled to less than 1 mm.

The prepared tissue samples were analysed for the following nutrient suite: calcium, magnesium, manganese, boron, copper, molybdenum, iron, zinc, sulfur, phosphorus and potassium, using nitric acid microwave digestion and elemental determination through inductively coupled plasma—optical emission spectrometry (ICP-OES). The nitrogen content in the tissue samples was determined using the Dumas method and LECO Trumac instrumentation (Lancrop Laboratories, personal communication).

The grain yield was determined using a plot combine (Deutz-Fahr M660), driven in a 2.1 m strip down the centerline of each plot, with grain samples taken during combining on 16th September 2022. In each plot four sub samples of approximately 100 g each, were manually collected from the combine at regular intervals, merged into a composite sample, air dried at 30˚C and stored in zip lock bags prior to shipping for analysis on 1st November 2022. The samples were analysed for the same nutrient and nitrogen suite as the tissue samples (see above), as well as for silica. Furthermore, the grain samples were analysed for the following PTEs; arsenic, cadmium, chromium, lead, mercury and nickel. PTEs and silica were

determined using a microwave-assisted aqua regia digestion and elemental determination through ICP-OES (Lancrop Laboratories, personal communication).

Plant tissue and grain samples were analysed using the United Kingdom Accreditation Service (UKAS) and ISO accredited (ISO/IEC 17025:2017) Lancrop Laboratories.

## Basalt

The basalt used in this study was from Divet Hill quarry (55.1003˚N, -2.03459˚W; Northumberland, UK). The composition of the basalt was determined using X-ray diffraction (XRD) for mineralogy, X-ray fluoresence (XRF) for oxide composition and ICP-OES and ICP-MS for elemental composition (Tables 1–3). For further basalt characterisation information see S2 Appendix. The basalt mineralogy is consistent with interpretations identified in [44], which identified products of hydrothermal alteration (quartz, calcite) and alteration of augite (to chlorite and smectite). In the absence of scanning electron microscopy-energy dispersive spectroscopy (SEM-EDS), it was not possible to determine the chemical composition of the amorphous phase(s). However, [44] states that talc occurs in association with pyroxene weathering, and this and other clay minerals, if poorly crystalline, may contribute to the amorphous component.

Chemical analyses (ICP-OES and XRF) data are consistent with each other (<0.42 wt.% difference) and XRF and XRD data are comparable to each other, when comparing mineral stoichiometry to both methods of chemical analysis. Using the TAS classification [17] and oxide data from XRF (Table 2), the Divet Hill material is classified as having a chemical composition of a basalt. The particle size ranges from 0–4 mm with a median particle size of 1147 μm (S2 Appendix).

**Table 1. Mineralogy of the Divet Hill basalt as determined using XRD.**

| Phase, identified from XRD | Mineral formula | Acid rate constant | Neutral rate constant | wt.% |
|---|---|---|---|---|
| *Faster dissolving silicate phases* | | | | |
| Andesine | $Na_{0.7-0.5}Ca_{0.3-0.5}Al_{1.3-1.5}Si_{2.7-2.5}O_8$ | -8.88 | -11.47 | 38.5 |
| Augite | $(Ca,Na)(Mg,Fe,Al,Ti)(Si,Al)_2O_6$ | -6.82 | -11.97 | 16.8 |
| Amphibole | $(Na,Ca,Fe,Mg,Al)_7Si_8O_{22}(OH)_2$ | -7.00[a] | -10.3[a] | 1.2 |
| *Medium dissolving silicate phases* | | | | |
| Chlorite + chlorite/smectite | $(Mg,Fe)_3(Si,Al)_4O_{10}(OH)_2 \cdot (Mg,Fe)_3(OH)_6$ | -11.11[b] | -12.52[b] | 7.8 |
| Illite+mica | $(K,Na,Ca)_2(Al,Mg,Fe)_{4-6}(Si,Al)_8O_{20}(OH,F)_4$ | -11.85[c] | -13.55[c] | 7.3 |
| *Slower dissolving silicate phases* | | | | |
| Quartz | $SiO_2$ | -13.40 | -16.29 | 11.4 |
| Smectite | $(Ca,Na)_{0.25-0.33}(Al,Mg,Fe)_{2-3}((Al,Si)_4O_{10})(OH)_2 \cdot nH_2O$ | -10.98 | -12.78 | 1.4 |
| *Non-silicate phases identified* | | | | |
| Calcite | $CaCO_3$ | -0.30 | -5.81 | 3.6 |
| Ilmenite | $FeTiO_3$ | -8.35 | -11.16 | 2.5 |
| *Un-known silicate phases* | | | | |
| Amorphous | - | - | - | 9.6 |
| Total | - | - | - | 100.1 |

[a] hornblende dissolution rate used for amphibole rates provided here

[b] chlorite dissolution rate constants used for chlorite + chlorite/smectite rates provided here

[c] muscovite dissolution rate used for illite+mica rates provided here

Proportions of each mineral provided is in wt.%. Where minerals form in solid solution, general mineral formulas are provided. Acid and neutral rate constants represent log(dissolution rate) at 25˚C. These constants are taken from [45].

**Table 2. Chemical composition of the Divet Hill basalt, as determined from XRF.**

| Compound | wt.% |
|---|---|
| $SiO_2$ | 50.02 |
| $Al_2O_3$ | 13.59 |
| $Fe_2O_3$ | 13.02 |
| CaO | 8.97 |
| MgO | 5.32 |
| $TiO_2$ | 2.35 |
| $Na_2O$ | 2.34 |
| $SO_3$ | 0.91 |
| $K_2O$ | 0.87 |
| $P_2O_5$ | 0.28 |
| MnO | 0.18 |
| BaO | 0.05 |
| SrO | 0.05 |
| $Cr_2O_3$ | 0.01 |
| Total | 97.96 |
| LOI | 2.17 |

The detection limit of all oxides is 0.01%.

## Statistical analysis

A linear mixed effect model (LMEM) was used to evaluate whether the effect of basalt treatment and cultivation type was statistically significant (following [46]). The treatment and cultivation factors were included as fixed effect terms and the block factor as the random effect term, the latter to account for the spatial heterogeneity between the blocks. Model assumptions regarding normality and homoscedasticity of variance of residuals were tested using Shapiro-Wilk [47] and White's Lagrange multiplier tests, respectively. Where model residuals do not meet LMEM assumptions, the data was Box Cox transformed to determine the significance of the fixed effects. The potential interaction effect of the fixed effects was investigated using a LMEM, but generally no interaction effects were found. For pH, the statistical analysis was performed on the measured pH values as well as the antilog of the negative pH value, using the untransformed hydrogen ion concentration as a continuous variable [48]. Model residuals were additionally assessed for normality and variance using kernal density estimate, quantile-quantile (Q-Q) and scatter plots. The effect of basalt amendment was also assessed using Wilcoxon and Kruskal-Wallis non-parametric tests. All statistical analyses were carried out using Python (version 3.9) and the package Statsmodels (version 0.13.2) [49].

## Results

### Grain yield

A significantly higher yield in the basalt-amended plots was observed across cultivation types (on average 20.5% and 9.3% for direct drill and ploughed plots, respectively), as well as a higher yield in the ploughed plots compared to the direct drill plots (Tables 4 and 5, Fig 2). The mean thousand grain weight in each group was notably similar (ranging from 41.14—41.59 g), indicating that the significant differences in grain yield were not due to larger or heavier grains, but rather a larger number of grains. It is also worth noting that a higher variance was observed within the direct drill plots compared to the ploughed plots (Fig 2).

**Table 3. ICP-OES and ICP-MS determination of the elemental composition of Divet Hill basalt.**

| Element | Unit | Analytical range | EU inorganic soil improver limits | Divet Hill |
|---------|------|------------------|-----------------------------------|------------|
| Ag | mg kg$^{-1}$ | 0.5–100 | | <0.5 |
| Al | % | 0.01–50 | | 7.06 |
| As | mg kg$^{-1}$ | 5–10000 | 40 | <5 |
| Ba | mg kg$^{-1}$ | 10–10000 | | 360 |
| Be | mg kg$^{-1}$ | 0.5–1000 | | 1 |
| Bi | mg kg$^{-1}$ | 2–10000 | | <2 |
| Ca | % | 0.01–50 | | 5.99 |
| Cd | mg kg$^{-1}$ | 0.5–1000 | 1.5 | 0.6 |
| Co | mg kg$^{-1}$ | 1–10000 | | 42 |
| Cr | mg kg$^{-1}$ | 1–10000 | 2 (Cr VI) | 60 |
| Cu | mg kg$^{-1}$ | 1–10000 | 300 | 66 |
| Fe | % | 0.01–50 | | 8.69 |
| Ga | mg kg$^{-1}$ | 10–10000 | | 20 |
| Hg | mg kg$^{-1}$ | - | 1 | 0.011 |
| K | % | 0.01–10 | | 0.72 |
| La | mg kg$^{-1}$ | 10–10000 | | 20 |
| Li | mg kg$^{-1}$ | 10–10000 | | - |
| Mg | % | 0.01–50 | | 2.99 |
| Mn | mg kg$^{-1}$ | 5–100000 | | 1295 |
| Mo | mg kg$^{-1}$ | 1–10000 | | 1 |
| Na | % | 0.01–10 | | 1.67 |
| Ni | mg kg$^{-1}$ | 1–10000 | 100 | 55 |
| P | mg kg$^{-1}$ | 10–10000 | | 1280 |
| Pb | mg kg$^{-1}$ | 2–10000 | 120 | 9 |
| S | % | - | | 0.37 |
| Si | mg kg$^{-1}$ | - | | 23.4 |
| Sb | mg kg$^{-1}$ | 5–10000 | | <5 |
| Sc | mg kg$^{-1}$ | 1–10000 | | 26 |
| Sr | mg kg$^{-1}$ | 1–10000 | | 391 |
| Th | mg kg$^{-1}$ | 20–10000 | | <20 |
| Ti | % | - | | 1.38 |
| Tl | mg kg$^{-1}$ | 10–10000 | | <10 |
| U | mg kg$^{-1}$ | 10–10000 | | <10 |
| V | mg kg$^{-1}$ | 1–10000 | | 300 |
| W | mg kg$^{-1}$ | 10–10000 | | <10 |
| Zn | mg kg$^{-1}$ | 2–10000 | 800 | 137 |
| Se | mg kg$^{-1}$ | - | | - |

All elements were determined using ICP-OES, except mercury which was determined using ICP-MS. EU inorganic soil improver limits are included for the elements where these are specified [30].

## Soil pH

A similar pattern to grain yield was seen in the soil pH data, where a significantly higher pH was observed in the basalt-amended soil, compared to the control plots, both for the direct drill (mean 6.76 and 6.47, respectively) and ploughed plots (mean 6.89 and 6.69, respectively) (Table 4). A significant difference was also found as a result of cultivation type, with a slightly higher pH in the ploughed plots (Fig 3, Tables 4 and 5).

**Table 4. The mean (M) and standard deviation (SD) of each determined parameter within each combination of treatment and cultivation type (n = 4).**

| | | Units | Control + Direct Drill | | Basalt + Direct Drill | | Control + Ploughed | | Basalt + Ploughed | |
|---|---|---|---|---|---|---|---|---|---|---|
| | | | *M* | *SD* | *M* | *SD* | *M* | *SD* | *M* | *SD* |
| Soil | pH | | 6.47 | 0.30 | 6.76 | 0.37 | 6.69 | 0.29 | 6.89 | 0.25 |
| Grain | Yield | t ha$^{-1}$ | 3.65 | 0.95 | 4.4 | 0.96 | 5.4 | 0.7 | 5.9 | 0.27 |
| | TGW | g | 41.51 | 1.67 | 41.59 | 1.64 | 41.81 | 2.05 | 41.14 | 2.37 |
| | Nitrogen | % | 2.23 | 0.02 | 2.10 | 0.15 | 2.00 | 0.10 | 1.99 | 0.07 |
| | Phosphorus | % | 0.40 | 0.04 | 0.41 | 0.04 | 0.37 | 0.03 | 0.38 | 0.04 |
| | Potassium | % | 0.42 | 0.04 | 0.44 | 0.04 | 0.39 | 0.03 | 0.42 | 0.03 |
| | Sulfur | % | 0.19 | 0.01 | 0.19 | 0.01 | 0.18 | 0.01 | 0.18 | 0.00 |
| | Calcium | % | 0.12 | 0.01 | 0.12 | 0.01 | 0.11 | 0.01 | 0.11 | 0.01 |
| | Magnesium | % | 0.14 | 0.01 | 0.14 | 0.01 | 0.13 | 0.01 | 0.14 | 0.01 |
| | Boron | mg kg$^{-1}$ | 0.96 | 0.11 | 1.02 | 0.14 | 0.87 | 0.12 | 0.98 | 0.05 |
| | Copper | mg kg$^{-1}$ | 5.88 | 0.22 | 5.45 | 0.39 | 4.95 | 0.31 | 5.10 | 0.24 |
| | Iron | mg kg$^{-1}$ | 57.05 | 4.68 | 58.97 | 6.61 | 57.98 | 3.54 | 57.83 | 2.71 |
| | Manganese | mg kg$^{-1}$ | 42.60 | 6.96 | 35.30 | 5.24 | 43.40 | 1.87 | 42.20 | 6.23 |
| | Molybdenum | mg kg$^{-1}$ | 0.15 | 0.09 | 0.23 | 0.08 | 0.14 | 0.03 | 0.17 | 0.08 |
| | Silica | mg kg$^{-1}$ | 181.50 | 32.44 | 218.00 | 12.19 | 206.50 | 37.60 | 201.25 | 30.50 |
| | Zinc | mg kg$^{-1}$ | 39.48 | 3.85 | 36.68 | 3.51 | 32.22 | 3.04 | 32.30 | 1.61 |
| Grain PTEs | Arsenic | mg kg$^{-1}$ | 0.08 | 0.08 | 0.07 | 0.06 | 0.08 | 0.08 | 0.16 | 0.08 |
| | Cadmium | mg kg$^{-1}$ | 0.05 | 0.01 | 0.05 | 0.01 | 0.03 | 0.01 | 0.04 | 0.01 |
| | Chromium | mg kg$^{-1}$ | 4.19 | 0.47 | 4.24 | 0.4 | 4.18 | 0.1 | 3.91 | 0.52 |
| | Nickel | mg kg$^{-1}$ | 2.97 | 0.33 | 2.84 | 0.22 | 3.03 | 0.45 | 2.82 | 0.21 |
| Tissue | Nitrogen | % | 2.7 | 0.16 | 2.42 | 0.05 | 1.92 | 0.16 | 2.02 | 0.22 |
| | Phosphorus | % | 0.14 | 0.03 | 0.13 | 0.01 | 0.12 | 0.01 | 0.12 | 0.01 |
| | Potassium | % | 1.26 | 0.14 | 1.38 | 0.18 | 1.57 | 0.27 | 1.71 | 0.11 |
| | Sulfur | % | 0.58 | 0.03 | 0.58 | 0.05 | 0.37 | 0.03 | 0.4 | 0.05 |
| | Calcium | % | 1.42 | 0.12 | 1.58 | 0.07 | 1.41 | 0.1 | 1.5 | 0.09 |
| | Magnesium | % | 0.24 | 0.01 | 0.23 | 0.02 | 0.19 | 0.03 | 0.21 | 0.03 |
| | Boron | mg kg$^{-1}$ | 11.13 | 1.08 | 11.83 | 0.19 | 9.9 | 1.1 | 10.6 | 0.34 |
| | Copper | mg kg$^{-1}$ | 5.95 | 0.62 | 5.3 | 0.14 | 4.03 | 0.46 | 4.05 | 0.26 |
| | Iron | mg kg$^{-1}$ | 96.25 | 6.85 | 87 | 6.38 | 88.25 | 9.64 | 81.5 | 4.65 |
| | Manganese | mg kg$^{-1}$ | 21.33 | 4.91 | 15.93 | 3.46 | 26.05 | 11.7 | 15.3 | 3.37 |
| | Molybdenum | mg kg$^{-1}$ | 0.51 | 0.09 | 0.66 | 0.19 | 0.63 | 0.21 | 0.54 | 0.2 |
| | Zinc | mg kg$^{-1}$ | 12.5 | 1 | 11.75 | 0.96 | 9.25 | 0.5 | 9.5 | 1.29 |

The grain samples were also analysed for the potentially toxic elements (PTEs) mercury and lead, but levels were below detection, and hence, data for these elements are not included in this table.

## Grain nutrients

Significantly higher grain nutrient concentrations were observed, as a result of basalt amendment for potassium and boron, whereas there is a significantly lower concentration of manganese (Table 5). Significantly lower grain concentrations from the ploughed plots compared to direct drill plots, were observed for nitrogen, phosphorus, potassium, sulfur, copper, boron and zinc (Table 4).

## Grain potentially toxic elements

No statistically significant accumulation of PTE concentrations was found as a result of basalt amendment (Table 5). Copper and zinc were analysed as part of the standard nutrient analysis,

**Table 5. p-values for each parameter, as derived from linear mixed effect modelling with cultivation and basalt amendment as fixed effects and the block factor set as a random effect.**

| | | Units | Cultivation | Basalt |
|---|---|---|---|---|
| Soil | pH | | **<0.001** | **<0.001** |
| Grain | Yield | t ha$^{-1}$ | **<0.001** | **0.015** |
| | TGW | g | *0.933* | *0.753* |
| | Nitrogen | % | **<0.001** | 0.135 |
| | Phosphorus | % | **0.005** | 0.320 |
| | Potassium | % | **0.008** | **0.008** |
| | Sulfur | % | ***<0.001*** | *0.670* |
| | Calcium | % | 0.059 | 0.637 |
| | Magnesium | % | 0.206 | 0.527 |
| | Boron | mg kg$^{-1}$ | **0.034** | **0.007** |
| | Copper | mg kg$^{-1}$ | **<0.001** | 0.402 |
| | Iron | mg kg$^{-1}$ | 0.960 | 0.692 |
| | Manganese | mg kg$^{-1}$ | 0.051 | **0.031** |
| | Molybdenum | mg kg$^{-1}$ | 0.276 | 0.073 |
| | Silica | mg kg$^{-1}$ | 0.733 | 0.195 |
| | Zinc | mg kg$^{-1}$ | **<0.001** | 0.300 |
| Grain PTEs | Arsenic | mg kg$^{-1}$ | 0.195 | 0.335 |
| | Cadmium | mg kg$^{-1}$ | **0.036** | 0.675 |
| | Chromium | mg kg$^{-1}$ | 0.250 | 0.455 |
| | Nickel | mg kg$^{-1}$ | 0.888 | 0.145 |
| Tissue | Nitrogen | % | **<0.001** | 0.330 |
| | Phosphorus | % | **0.002** | 0.255 |
| | Potassium | % | **<0.001** | **0.014** |
| | Sulfur | % | **<0.001** | 0.417 |
| | Calcium | % | 0.175 | **<0.001** |
| | Magnesium | % | **<0.001** | 0.639 |
| | Boron | mg kg$^{-1}$ | **<0.001** | **0.026** |
| | Copper | mg kg$^{-1}$ | **<0.001** | 0.126 |
| | Iron | mg kg$^{-1}$ | **0.025** | **0.008** |
| | Manganese | mg kg$^{-1}$ | 0.700 | **<0.001** |
| | Molybdenum | mg kg$^{-1}$ | 0.973 | 0.660 |
| | Zinc | mg kg$^{-1}$ | **<0.001** | 0.610 |

Bold numbers show a significant effect of the fixed factors at a 95% confidence level. Italicised numbers are for parameters where model residuals were not normally distributed (Shapiro-Wilk p-value <0.05), the models did converge and there was homogeneity in the variance of the residuals. Tissue manganese and phosphorus were BoxCox transformed to satisfy normality and homoscedasticity of model residuals.

as well as part of the PTE suite, but only reported under nutrients, as the concentrations are far below toxicity (Table 4). A marginally lower uptake of cadmium was observed as a result of ploughing (p<0.05, Tables 4 and 5). Lead and mercury are not reported as concentrations were below the detection limit (0.5 mg kg-1).

## Tissue nutrients

Significantly higher tissue concentrations of calcium, potassium and boron were found in the basalt-amended plots (Table 4), whereas significantly lower concentrations of manganese and iron were found. Similar to the grain nutrient concentrations, all nutrients (nitrogen, phosphorus, sulfur, magnesium, iron, copper, boron and zinc) except potassium were significantly

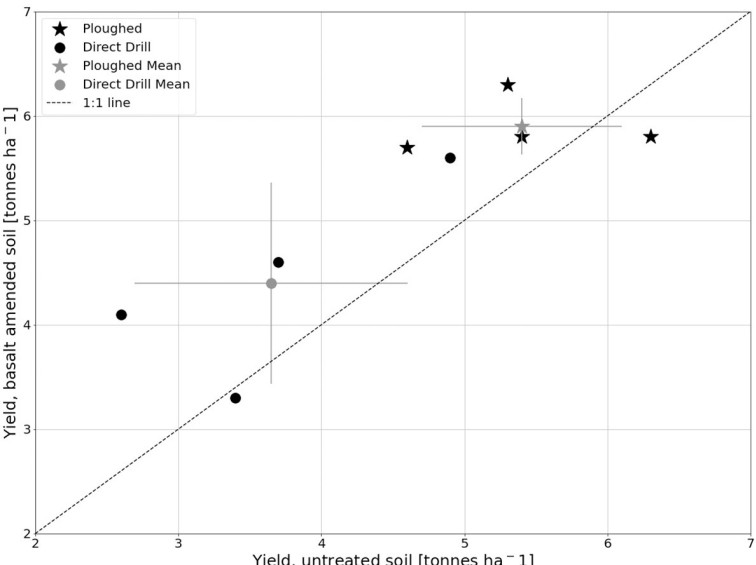

**Fig 2. Yield data.** Yield from basalt-amended plots plotted against yield from untreated plots, paired using cultivation and block number. Mean yield from both cultivation types is visualised with the standard deviation as error bars. Dotted line indicates the 1:1 line.

higher in plant tissues in the direct drill plots relative to the ploughed treatments (Tables 4 and 5).

## Discussion

The aim of this study was to investigate the first-year effects of applying crushed basalt to agricultural fields. The study focuses on yield responses of spring oat, tissue and grain

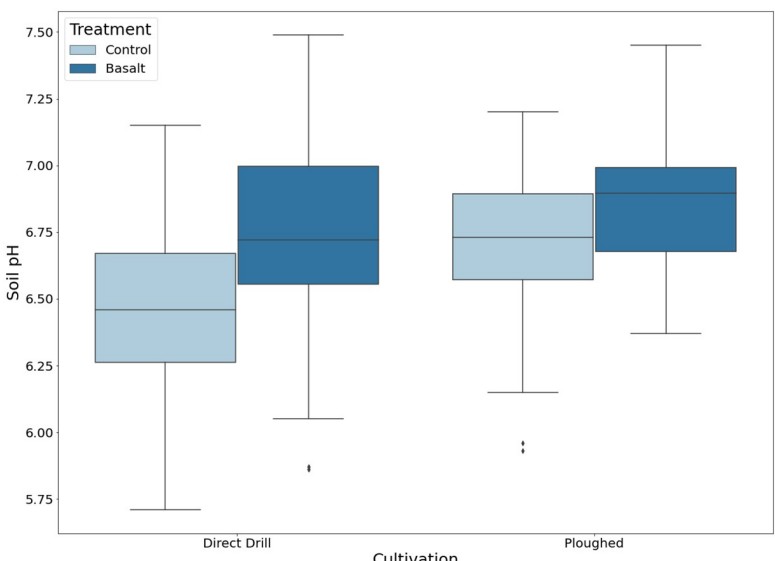

**Fig 3. Soil pH data.** Box plots of soil pH from basalt-amended and control plots for each cultivation type (n = 64). Boxes depict the quartile of each group, whiskers show the extent of the distribution, diamond points have been classified as outliers. The pH was determined in a 1:2.5 soil:$H_2O$ ratio.

nutrient and PTE uptake, as well as differences in soil pH between basalt-amended and control plots.

## Yield

Basalt amendment showed a significantly higher crop yield in both ploughed and direct-drilled plots (Table 5, Fig 2). Concurrently, the average mass of one thousand grains across all four groups did not vary, ranging between 41.14 and 41.81 g (Table 4), indicating that the higher yield was not due to larger or heavier grain, but rather a higher abundance of grain.

The difference in mean yield due to basalt amendment on the direct drill plots was 0.75 tonnes ha$^{-1}$ (i.e. 4.40 vs. 3.65 tonnes ha$^{-1}$) and on the ploughed plots 0.50 tonnes ha$^{-1}$ (i.e. 5.90 vs. 5.40 tonnes ha$^{-1}$), see Table 4. This corresponds to a mean difference of 20.5 and 9.3%, respectively, although it should be noted that the variance within each group was relatively large (with standard deviations ranging from 0.27–0.95 tonnes ha$^{-1}$, Table 4).

First-year differences in yield were also observed for potato and maize in a study in southern Denmark [22], although the median particle size of the glacial rock flour used in that study is considerably smaller than the basalt material applied in this study (2.6 μm vs. 1147 μm). A similar average increase of 21% in *Sorghum bicolor* yield was found in a controlled mesocosm-scale experiment with an application rate of 100 tonnes ha$^{-1}$ (a fivefold increase in basalt application rate compared to this study), sufficient irrigation and controlled temperatures [15]. An even greater increase of 100% in soybean yield was reported in a Canadian microplot experiment [50], with the magnitude of this increase potentially explained by the use of the faster-dissolving mineral wollastonite, again delivered at a relatively high application rate of 100 tonnes ha$^{-1}$.

The findings in this study are noteworthy because the experimental boundary conditions were expected to lead to lower rates of dissolution compared to the studies discussed above, and hence, smaller crop yield responses. The current study was limited by a basalt particle size up to 4 mm, a reduced application rate, a relatively dry growing season (S1 Fig) and minimal basalt incorporation into the soil. Hence, the magnitude of the difference in yield observed here as a result of basalt amendment were initially unexpected during the first growing season. However, given the dry growing season conditions, crops grown in basalt-amended plots may have been bolstered by improved drought resistance conferred through the uptake of additional silicon released from fast-dissolving silicate minerals [51–53]. Whether significantly higher yields of this magnitude due to basalt amendment would have occurred in a growing season that was not relatively dry will need to be studied in future trials. However, it may be an indication that soil amendments using crushed basalt have the potential to improve crop resistance to drier conditions.

Statistically, the strongest effect on crop yield was that of cultivation type (p<0.005, Table 5), with a difference in the mean yield across basalt amendment of approximately 2 tonnes ha$^{-1}$ (Fig 2). However, the variance within the direct drill groups, for both the control and basalt-amended treatments, was much higher than in the ploughed groups (Table 4). It is possible that poor establishment in the direct drill plots gave rise to this result, with an uneven and patchy crop stand observed throughout the growing season. The direct drill cultivation was introduced in 2019 and drilling dates and seeding rates were kept constant with the ploughed plots henceforth. These factors can be expected to result in poorer performance, particularly on heavy clay soils in temperate climates where yield depression after transitioning to direct drilling methods is common (especially in the initial years after switching from ploughing [54]), and often commercially compensated for by varying drilling dates and increasing seeding rates. This may also explain why the findings here contradict those of other studies,

where substantial increases in crop yield and aboveground biomass have been obtained from direct drill fields [55]. Soil moisture conditions at the time of drilling may have also been unfavourable in the direct drill plots, leading to the high variability observed.

No interaction effect was found between the cultivation types and basalt amendment for yield, this being similar to all other parameters modelled. The lack of any interaction effect could be attributed to the limited incorporation of basalt during seeding, given the target depth of 2 cm for both ploughed and direct drill plots; i.e. the basalt was applied after tillage and not incorporated mechanically into the soil on the ploughed plots. Hence, in this study, the two cultivation types are not representing different basalt incorporation strategies, but rather differences in the underlying soil structure, for which the effect of the basalt amendment appears agnostic.

## Soil pH

Basalt amendment led to a significantly higher soil pH relative to the control plot, in both the direct drill (on average 6.47 vs. 6.76) and the ploughed plots (on average 6.69 vs. 6.89) (Table 4). Hence, a moderate, but significantly higher pH (on average 0.29 and 0.2 pH units) was observed 256 days after basalt application. Although soil samples were not collected during the growing season, it can be surmised that the significant difference in yield seen with basalt application was probably driven by modest differences in soil pH that manifested during the crop cycle. In support, indications of differences in pH between the plots during the growing season were observed in the significantly reduced uptake of manganese and iron in the basalt-amended plots (Tables 4 and 5). The plant availability of these nutrients decreases abruptly above pH 6.5 [4, 56, 57]. Although care should be taken when assessing the absolute pH levels, the pH of the soil in this study varied around pH 6.5 (6–7.5, Fig 3).

The mean difference in soil pH seen here is in the same order of magnitude as that observed in a study using a rock containing the faster dissolving mineral, wollastonite (a mineral that dissolves up to 1200 times faster than faster weathering silicate minerals [45]), applied at 30 tonnes ha$^{-1}$. The mean difference reported in [50] was 0.3 pH units compared to the control soil, which was achieved after only 98 days. Furthermore, the crops used in this study were nitrogen fixing soybean and alfalfa [50], that have a stronger acidifying effect on the soil, compared to spring oat. Hence, the effect on soil pH may have been lower in the present study had a similar N-fixing crop been used. The starting pH of the soil in [50] was similar to the circum-neutral pH of the soil in this study. In more acidic soils, even larger changes in soil pH have been observed, e.g. [58] reported a change of 0.5 pH units (from 4.7 to 5.2) within 9 months of application of 20 tonnes ha$^{-1}$ basalt. However, dissolution rates would be expected to be more favourable in the study by [58], where the particle size used was less than 250 μm, and the study was conducted in a Malaysian cocoa plantation with tropical temperatures and precipitation rates.

Compared to the above studies, the boundary conditions in this study are less favourable, and so the difference in soil pH between the control and basalt-amended plots were expected to be smaller within the first year of application. The significant first-year differences in pH may be ascribed to the 3.6 wt.% content of faster dissolving calcite (Table 1). The calcite (calcium carbonate) fraction contribution in this study was 0.68 tonnes ha$^{-1}$. Comparatively, if a limestone product with a 70% effective calcium carbonate equivalent (similar to a limestone product used in [59]) and an application density of 2 to 5 tonnes ha$^{-1}$ (typical single application rate in NE England, depending on the soil texture and starting pH) had been used, the contribution would range between 1.4 to 3.5 tonnes ha$^{-1}$ calcium carbonate (between two and fivefold higher concentrations than that of the basalt). A study of six basalts from different

geological provinces identified carbonate concentrations up to 1.25 wt.% [60], compared to which the content in the Divet Hill feedstock is relatively high. However, as illustrated above, despite the relatively high calcite content in the Divet Hill basalt, a single application with a target application rate of 20 tonnes ha$^{-1}$ would not be sufficient to replace a typical single lime application.

The faster dissolving silicate minerals andesine, augite, amphibole and possibly the amorphous phase are also likely to have contributed to the difference in soil pH, although the neutral dissolution rates (which would be dominant under the soil pH conditions in this study) are at least 30,000 times slower than the dissolution rate for calcite (Table 1). Dissolution of exposed carbonate minerals can rapidly elevate soil pH in the short term, whereas the weathering of silicate minerals will have a longer-term neutralising effect on soil pH, due to comparatively slower dissolution rates (Table 1). Hence, basalts containing calcite may be attractive for producers looking to quickly modify the pH of their soil. The available dataset does not permit quantification of the relative contribution of calcite and silicate mineral dissolution to the observed pH effect.

A significantly lower pH in the direct drill plots relative to the ploughed plots was also observed (Table 5). This difference may be due to increased conservation of soil organic matter in the direct drill plots since implementation of this cultivation type in 2019 [61], although this was not investigated in the current study.

## Nutrient uptake

First-year differences in nutrient uptake include a significantly higher concentration of potassium in both grain and plant tissues and a higher calcium concentration in plant tissues, in samples taken from the basalt-amended plots. The significantly higher grain potassium concentration supports results obtained by [15], whereas the higher tissue concentration of potassium was not observed by [15]. The significantly higher concentration of potassium, both in tissue and grain, across cultivation types is not likely to be solely an indication of changes to soil pH, as the availability of potassium is not believed to increase markedly with changes in pH within the ranges observed in this study [57, 62]. This is also in line with [63] who observed a lower decrease in soil potassium in basalt-amended plots, compared to lime amended plots, where soil pH was elevated. It is therefore hypothesised that the elevated potassium observed in tissue and grain may have originated from other processes in addition to the differences in soil pH.

The additional potassium may have originated from a variety of sources including: (1) the rock, (2) the soil and (3) via symbiotic plant-microbe relationships. Considering the rock, potassium may have originated from the illite/mica phase identified through XRD, as potassium is a key component of illite and several mica minerals (e.g. muscovite and biotite). This phase is generally considered to have a medium-fast dissolution rate, but will still dissolve and may release additional potassium into soil pore water. Additionally, similar to calcium and magnesium, potassium within the illite/mica phase may be preferentially released into the soil solution before the silicon-oxygen bonds are broken, as hypothesised [64] and observed [65] in previous studies. Considering the soil, calcium and magnesium released from faster weathering minerals can displace native potassium from soil cation exchange sites, making it more accessible to plants. Competitive adsorption of this kind was observed in [27], where the EW feedstock used was olivine, which did not contain potassium. Additionally, potassium may have also been preferentially mined from the mineral structure by microorganisms, including mycorrhizal fungi [66], though this hypothesis was not investigated in this study.

A significantly higher calcium uptake in plant tissue from the basalt-amended plots was observed across cultivation types, whereas no such difference was observed for magnesium. Both calcium and magnesium are readily weathered from basalt, and are thought to be preferentially leached from minerals before the breakdown of silicon-oxygen bonds within minerals [64]. The Divet Hill basalt contains calcium (CaO: 8.97 wt.%) and magnesium (MgO: 5.32 wt. %) (Table 2), and both cations are present in the faster-dissolving silicate phases (e.g. augite and amphibole—Table 1). However, calcium is also present in calcite (Table 1—which dissolves at least 30,000-fold faster than augite and amphibole), and is posited here to be the main source of calcium during the first growing season. In a tropical mesocosm-scale study, using a basalt with similar chemical composition as Divet Hill (CaO: 8.92 wt.% and MgO: 5.36 wt.%) and a sandy clay loam soil, a significant increase in both calcium and magnesium content of plant shoots was observed in maize and soy [59], suggesting that magnesium may also be provided through basalt application. In a later, similar study from the same research group [67] no increases in soil magnesium were detected, only soil calcium, as a result of basalt amendment. Although significant differences in magnesium uptake were not evidenced during the first growing season in the current study, it may manifest during following growing seasons, as the absence of elevated magnesium plant uptake is not necessarily due to the absence of basalt dissolution. [15] found no magnesium increase in plant roots and shoots, but showed a significant decrease in magnesium in basalt grains extracted from the experiment (compared to unweathered basalt grains). Further, in [15] there was a significant increase in magnesium on soil cation exchange surfaces (as interpreted from ammonium acetate-extractions), which suggests that some basalt weathering products may be less available to plants than others.

Boron levels, in both grain and plant tissue, were also significantly higher in response to basalt amendment of soils in the current study. Reasons for this remain unclear and warrant further study.

The essential plant nutrient, phosphorus, is present in the Divet Hill basalt at concentrations of 0.1280 wt.% (Table 3) although we did not detect any calcium phosphate apatite in the rock. At this concentration of phosphorus, apatite would be present at a concentration of 0.7 wt.%, which is not likely to be resolved using XRD. Furthermore, we did not identify a significant difference in the phosphorus concentration in the plant grain or tissue as a result of basalt amendment (Table 5).

Where significant differences were observed in grain and tissue analysis between the two cultivation types (Table 5), the nutrient concentrations in the direct drill plots were always higher, with the sole exception of tissue potassium (Table 4). Although direct drill systems may provide advantages for retaining soil nutrients, this elevated uptake could be related to the poor crop stand observed in these plots, resulting in reduced competition for nutrients with neighbouring plants.

## Potentially toxic elements

There was no evidence that application of crushed basalt rock led to accumulation of any PTEs in grain, suggesting that crops amended with this product are safe for human and animal consumption (Tables 4 and 5). Product toxicity was not expected in the current study, particularly given that element concentrations in the Divet Hill basalt do not exceed any of the EU thresholds for inorganic soil improver products (Table 3). Dietary limits for PTEs in cereals have been identified for cadmium (0.1 mg kg$^{-1}$) and lead (0.2 mg kg$^{-1}$), mercury in fishery products (0.5 mg kg$^{-1}$, [68]) and for arsenic in wheat (1 mg kg$^{-1}$, [69]). The grain concentrations in this study were well below these limits, with lead and mercury being undetectable and maximum concentrations for cadmium and arsenic of 0.07 mg kg$^{-1}$ and 0.26 mg kg$^{-1}$, respectively. While

there are no published dietary limits for chromium and nickel, it is reassuring that no significant increases in grain concentrations were observed for these elements (Table 5).

## Conclusion

The application of crushed basalt rock resulted in significantly higher spring oat yields across both direct drill and ploughed plots, during the first growing season. The difference in yield between the control and basalt-amended plots is primarily ascribed to a modest difference in soil pH, which during the growing season led to reduced grain and tissue manganese uptake, as well as reduced tissue iron uptake. The reported soil pH differences were likely primarily due to the rapid dissolution of calcite, and to a less extent, the dissolution of faster-dissolving silicate minerals. The former is supported by significantly higher tissue calcium content, whereas the latter is supported by a significantly higher grain and tissue potassium, in the crops from the basalt-amended plots. However, it is important to note that the exceptionally dry 2022 growing season in NE England may have amplified crop yield differences, through secondary factors that were not fully examined by this study. The significant differences that arose in this study due to basalt amendment were ostensibly independent of cultivation type (effectively the underlying soil structure), probably because of the lack of mechanical incorporation of basalt into the ploughed soil. Differences in yield and plant nutrients due to cultivation are largely ascribed to poor emergence (and thus lower competition for nutrients between plants) of spring oat grown in the direct drill plots. Basalt application did not result in the accumulation of PTEs in grains compared to the control plots, indicating that crops amended with this particular crushed basalt rock are safe for consumption.

## Supporting information

**S1 Fig. Weather data.**
(PDF)

**S1 Appendix. Herbicide and fungicide applications.**
(PDF)

**S2 Appendix. Basalt characterisation.**
(PDF)

## Acknowledgments

The authors would like to thank all the employees at the Nafferton research farm that were involved in this work, including (but not limited to) James Standen, Gavin Hall and Rachel Chapman.

## Author Contributions

**Conceptualization:** Julia Cooper, Dave George, David Manning, Yit A. Teh.

**Data curation:** Kirstine Skov, Jez Wardman.

**Formal analysis:** Kirstine Skov.

**Funding acquisition:** Jim Mann.

**Investigation:** Jez Wardman.

**Methodology:** David Manning, Yit A. Teh.

**Project administration:** Xinran Liu.

**Resources:** Jim Mann.

**Supervision:** Yit A. Teh, Xinran Liu.

**Validation:** Jez Wardman, Matthew Healey, Amy McBride, Tzara Bierowiec, Ifeoma Edeh, Mike E. Kelland, Melissa J. Murphy, Ryan Pape, Will Turner, Peter Wade, Xinran Liu.

**Visualization:** Kirstine Skov.

**Writing – original draft:** Kirstine Skov, Amy McBride.

**Writing – review & editing:** Kirstine Skov, Jez Wardman, Matthew Healey, Amy McBride, Tzara Bierowiec, Julia Cooper, Ifeoma Edeh, Dave George, Mike E. Kelland, David Manning, Melissa J. Murphy, Ryan Pape, Yit A. Teh, Will Turner, Peter Wade, Xinran Liu.

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
