## [Decision Letter · Decision Letter 0]

16 Jun 2023

PONE-D-23-15404

Initial agronomic benefits of enhanced weathering using basalt: A study of spring oat in a temperate climate

PLOS ONE

Dear Dr. Skov,

Thank you for submitting your manuscript to PLOS ONE. Based on the reviews recieved and my own assessment as the Academic Editor, I feel that your manuscript has merit but as you will appreciate herein, the reviewers have raised major concerns that means it does not fully meet PLOS ONE’s publication criteria as it is. Therefore, I invite you to submit a revised version of the manuscript that addresses the points raised during the review process.

We look forward to receiving your revised manuscript.

Kind regards,

Timothy Omara; BSc, MSc, PhDc

Academic Editor

PLOS ONE

Journal Requirements:

   "The study design and yield data collection was conducted by Newcastle University, independent of funders of the project.

All other data collection and analysis was funded by UNDO Carbon ltd.

Decision to publish and preparation of the manuscript was done in collaboration between Newcastle University and UNDO Carbon ltd."

   "Kirstine Skov, Jez Wardman, Matthew Healey, Amy Lewis, Tzara Bierowiec, Ifeoma Edeh, Melissa J. Murphy, Ryan Pape, Will Turner, Peter Wade and Xinran Liu are presently or were recently employed in the Science and Research department and may hold options in UNDO Carbon ltd.

Mike Kelland is an independent consultant for UNDO Carbon ltd. 

Jim Mann is the founder and CEO of UNDO Carbon ltd.

David Manning is part of UNDO carbon ltd.s scientific advisory board.

Julia Cooper, Dave George and Yit A. Teh declare no conflicts of interest."

6. We note that you have stated that you will provide repository information for your data at acceptance. Should your manuscript be accepted for publication, we will hold it until you provide the relevant accession numbers or DOIs necessary to access your data. If you wish to make changes to your Data Availability statement, please describe these changes in your cover letter and we will update your Data Availability statement to reflect the information you provide."

7. We note that you have included the phrase “data not shown” in your manuscript. Unfortunately, this does not meet our data sharing requirements. PLOS does not permit references to inaccessible data. We require that authors provide all relevant data within the paper, Supporting Information files, or in an acceptable, public repository. Please add a citation to support this phrase or upload the data that corresponds with these findings to a stable repository (such as Figshare or Dryad) and provide and URLs, DOIs, or accession numbers that may be used to access these data. Or, if the data are not a core part of the research being presented in your study, we ask that you remove the phrase that refers to these data.

8. We note that Figure 1 in your submission contain map/satellite images which may be copyrighted. All PLOS content is published under the Creative Commons Attribution License (CC BY 4.0), which means that the manuscript, images, and Supporting Information files will be freely available online, and any third party is permitted to access, download, copy, distribute, and use these materials in any way, even commercially, with proper attribution. For these reasons, we cannot publish previously copyrighted maps or satellite images created using proprietary data, such as Google software (Google Maps, Street View, and Earth). For more information, see our copyright guidelines: http://journals.plos.org/plosone/s/licenses-and-copyright.

Academic Editor's Comments:

Please avail the details requested by reviewers (as much as possible). A closer look at the manuscript suggests that most of the items provided as supplementary files should have been supplied in the main manuscript file to provide a better understanding of the study. Please consider this while performing your revision. 

Reviewers' comments:

Reviewer's Responses to Questions

**Comments to the Author**

1. Is the manuscript technically sound, and do the data support the conclusions?

Reviewer #1: Partly

Reviewer #2: Yes

2. Has the statistical analysis been performed appropriately and rigorously? 

Reviewer #1: I Don't Know

Reviewer #2: Yes

3. Have the authors made all data underlying the findings in their manuscript fully available?

Reviewer #1: Yes

Reviewer #2: Yes

4. Is the manuscript presented in an intelligible fashion and written in standard English?

Reviewer #1: Yes

Reviewer #2: Yes

5. Review Comments to the Author

Reviewer #1: This study provides one of the first field trial results of basalt rock use for enhanced weathering (EW) in the agricultural context. It is a short-term small-plot studies, so not quite a true "field study", but still has some value in taking the research forward from conceptual to deployment. The study does have some shortcomings in terms of data collected and results interpretation, and below these are discussed with suggestions for improvement and re-consideration.

Lines 11-12: interesting to cite such old studies on the suggestion of basalt as a soil mineral. What happened that this pratice has not become widespread? Can a bit more about the history of basalt as a soil amendment be said to explain the current state and why CDR is needed as an added incentive?

Line 56: it says "basalt does not necessarily contain the mineral olivine". Coincidently, I just spoke to a geologist yesterday who said the opposite, that some basalts have olivine. The compositional variation of basalt is something concerning about several studies referring to basalt as if being a well defined material. Can a bit more discussion be added to the introduction to contrast how variable basalt can be, and what makes a basalt suitable for the intended application?

Lines 60-72: this paragraph seems a bit disconnected to the rest of the introduction. It can be put into better context with connecting start and end sentences.

Line 76: the term "grant access to their land to apply it" suggests that EW is a technology in the hand of third parties who seek permission of landowners to apply, which is kind of how the MRV market has been shaping up... Wouldn't it be better for EW to be a technology that farmers want to implement in their fields out of their own interest? If there agronomic benefits, and agronomic role, then agronomic practices such as amendment rates based on standardized recommendations and amendment methods that are optimized for the crop should be used, rather than for maximizing carbon removal certificates. Is this the direction that the authors are looking into in terms of justifying basalt use based on agronomic benefits?

Line 105: how was the amendment rate of 18.86 tonnes per hectare selected? It is a reasonable rate (i.e., not as high as some studies suggest), so it would be especially useful to know what the rationalle was to better guide future studies.

Lines 105 to 108: it is confusing. If the plots were ploughed at the end of March, how was the basalt incorporated into the soil upon broadcast spreading? Not clear how the mineral was mixed into the soil, was it lightly tilled after spreading?

Line 126: <2 mm?

Line 128: what were the SOC and SIC contents of the soil?

Table 1 and Table S2: in how much agreement are the mineral composition and the chemical composition, stoichiometry-wise?

Table 1: "fast" is relative, is a term like "faster" more appropriate here?

Table 1: as this article is about the agronomic benefits of basalt, and part of those agronomic benefits would come from the release of P and K plant nutrients (from Table S2), the question is where are those elements present in the mineralogy? In addition, what is the mineral assemblage of these phases in the particles? SEM-EDX or EPMA study of basalt is something largely lacking from basalt studies. It is not clear which of the mineral phases would react first, under what extent of weathering it could be expected that the plant benefiting elements would be released, or even what secondary minerals may form during the weathering sequence that could passivate the remaining minerals from further weathering. Considering that this is a single-growth season study, it is even more important to understand what part of basalt can react within a few months.

Figure 2: interesting that one plot each of direct drill and ploughed (i.e., one circle and one star) are below the 1:1 line. Is there any spatial relationship (looking at Figure 1) in the plots the performed better and performed worse? For instance, the blue plots of block 1 are closer to trees, and trees tend to absorb nutrients from neighboring crops (in croplands, plants at the edge near tree lines always grow less). Also, is there any sense about the direction of run-off across the field? Also, was water infiltration rates equal in each block?

Figure 3: considering that the box plots have rather large whiskers, three boxes that are very much overlapping, and even some outlier points, it is surprising that Table 3 reports p values <0.001 for both cultivation and basalt effects. Likewise so many p values in Table 3 are <0.001 yet the values in Table 2 are not very different between the plots. Here also the question about the effect of spatial variability versus treatment effect comes in.

General comment: the only sign of weathering presented is the pH effect. This being an EW study, why were no measurements that are under consideration for MRV reported, such as: calcimetry, soil TGA, soil XRD, soil water alkalinity, soil water electrical conductivity, soil CO2 gas, soil isotope ratios, etc.

Discussion section: how much of the effect of basalt amendment on yield is due to the pH effect versus other EW effects? Considering how slowly basalt has been reported to weather, it can be expected that the weathering extent in the study period was small. So other than buffering soil pH, how can it be deemed that basalt in this study performed better than conventional liming with limestone? Plants tend to have higher yield and better nutrient uptake when grown in soils under optimal pH, so could this largely explain the results reported? If not, then how to make a conclusive case that basalt's effects were more than just its liming effect? In hindsight, this study would have benefited from an extra set of plots with conventional liming used.

Lines 243 to 245: This statement highlights how the results are somewhat surprising. This calls for a more definitive understanding as to why the results were better than expected. The basalt used was certainly rather coarse, which would suggest even slower weathering than other studies have suggested. So why were the results so good? Perhaps part of the "amorphous" content of the mineral was made up of free alkaline oxides that reacted quickly? Perhaps this andesine-rich basalt (which from literature seems to not be so common) is more reactive than typical basalts? Is the rock used actually basaltic andesite rather than basalt? Some of this is commented in later lines, but without a definitive conclusion other than the suggestion that the basalt used may have been more reactive than other basalts.

Line 274: here is more information about the basalt amendment that lacked earlier. Still not clear if the basalt was evenly tilled into the soil during this seeding process.

Lines 293 to 294: this statement about the reference is not correct. The cited study states: "At the end of the growth trial, the pH of the MSA 7.5 and MSA 10 soils was 7.43 ± 0.04 and 7.76 ± 0.05, respectively, whereas for the MSA 5 soil the pH was 7.11 ± 0.03 (Figure S6A)."

Line 337: but there is no evidence in this study of what these potassium-bearing basaltic phases are... If the liming effect is due to calcite, as suggested earlier, than how is a significant amount of K being released? Also back to the statistics, looking at Table 2 it is hard to believe that the K values are statistically different. In the grain, the values are 0.42 +/- 0.04 versus 0.44 +/- 0.04 for direct drill, which are very much overlapping values, and 0.39 +/- 0.03 versus 0.42 +/- 0.03 for ploughed, which also are perfectly overlapping values. Lines 338 to 346 attempt to explain possible mechanisms, but very hypothetically. The odds that other soil effects are responsible for the small changes in K uptake by plants, such as cation exchange with innate K in soils, are likely larger than to attribute to the small odds that poorly weathered minerals incongruently released K in soils. If this was the case, soil remineralization would already have overtaken the use of potash in many countries that have a hard time affording to import potash.

General comment: it is commented in the text that basalt may have impacted draught resistance. One effect that mineral amendments can have is in improving soil texture, and hence improving water infiltration and retention near the plant roots. In another basalt study that I have reviewed, that seemed to largely explain an effect that was claimed. Perhaps this effect should be considered in hypothesizing reasons for the short-term effects seen.

Conclusions: based on the aforementioned comments, some of the conclusion statements could be reconsidered. The conclusion does speak to the pH effect being a large effect, and does pose that calcite rather than silicate dissolution may explain much of the results. Still, there are comments about potassium that seems to be stretching the certainty of the results, and the last statement that "crops amended with crushed basalt are safe for consumption" is far too over-reaching considering how diverse basalt composition can be. This type of statement can be taken very much out of context in the MRV world. The conclusion section, which many reader read more than the rest of the text, should be more balanced to inform the reader than this was a short-term experiment under less than ideal field conditions and that many of the results are not well understood as there is no supporting mechanistic evidence provided in this study being the comparative analyses that are prone to variability effects and effects not considered in the study.

Reviewer #2: This manuscript has discussed the effect of applying basalt powder to the soil, leaf tissue and grains of oat plants in a temperate climate. This information is important for the academic community and especially for farmers. The manuscript is well written and its discussions are well reasoned. Authors can find some suggestions/annotations in the attached file. I recommend the approval of the manuscript if the due suggestions are duly answered or justified.

6. PLOS authors have the option to publish the peer review history of their article (what does this mean?). If published, this will include your full peer review and any attached files.

Reviewer #1: **Yes: **Rafael M. Santos

Reviewer #2: No

---

## [Author Response · Author response to Decision Letter 0]

26 Sep 2023

Response to reviewer comments can be found in the file ResponseToReviewers.pdf

---

## [Decision Letter · Decision Letter 1]

15 Oct 2023

PONE-D-23-15404R1Initial agronomic benefits of enhanced weathering using basalt: A study of spring oat in a temperate climatePLOS ONE

Dear Dr. Skov,

Thank you for submitting your manuscript to PLOS ONE. After careful consideration, we feel that it has merit but does not fully meet PLOS ONE’s publication criteria as it currently stands. Therefore, we invite you to submit a revised version of the manuscript that addresses the points raised during the review process.

We look forward to receiving your revised manuscript.

Kind regards,

Timothy Omara, PhD

Academic Editor

PLOS ONE

Journal Requirements:

Reviewers' comments:

Reviewer's Responses to Questions

**Comments to the Author**

1. If the authors have adequately addressed your comments raised in a previous round of review and you feel that this manuscript is now acceptable for publication, you may indicate that here to bypass the “Comments to the Author” section, enter your conflict of interest statement in the “Confidential to Editor” section, and submit your "Accept" recommendation.

Reviewer #1: (No Response)

Reviewer #2: All comments have been addressed

2. Is the manuscript technically sound, and do the data support the conclusions?

Reviewer #1: Partly

Reviewer #2: Yes

3. Has the statistical analysis been performed appropriately and rigorously? 

Reviewer #1: Yes

Reviewer #2: Yes

4. Have the authors made all data underlying the findings in their manuscript fully available?

Reviewer #1: Yes

Reviewer #2: Yes

5. Is the manuscript presented in an intelligible fashion and written in standard English?

Reviewer #1: Yes

Reviewer #2: Yes

6. Review Comments to the Author

Reviewer #1: The authors have made useful revisions, though there still a couple of topics to improve:

A) Soil pH results:

1) The manuscript says: "The mean difference in soil pH seen here is in the same order of magnitude as that observed in a study using a rock containing the faster dissolving mineral, wollastonite (a mineral that dissolves up to 1200 times faster than faster weathering silicate minerals [45]), applied at 30 tonnes ha−1 (mean difference of 0.3 pH units to the control soil after 98 days [50]). The starting pH of the soil in [50] was similar to the circumneutral pH of the soil in this study."

The study [50] used soybean and alfalfa, which are both nitrogen-fixing plants, which therefore have an acidifying effect during plant growth. In the present study of the authors, the plant used was oat, which is not a nitrogen-fixing plant, and therefore does not cause soil acidification to the same extent. So a pH elevation in a soil where the plant is countering the liming effect is much harder to achieve. This can explain why a similar amendment rate of the much faster weathering silicate could have had a similar pH change effect to basalt. So the direct comparison should be done more caustiously.

2) The manuscript says: "The significant first year differences in pH may be ascribed to the 3.6 wt.% content of faster dissolving calcite (Table 1). The calcite (calcium carbonate) fraction contribution in this study was 0.68 tonnes ha−1. Comparatively, if a limestone product with a 70% effective calcium carbonate equivalent and an application density of 2 to 5 tonnes ha−1 (similar to a limestone product used in [59] had been used, the contribution would range between 1.4 to 3.5 tonnes ha−1 calcium carbonate (between two and fivefold higher concentrations than that of the basalt). A study of six basalts from widely different geological provinces found that basalts contain up to 1.25 wt.% carbonates [60], compared to which the content in Divet Hill is relatively high."

The study [59] was comparing limestone applications between 1 to 4 tonnes/hectare to basalt application of 33 to 99 tonnes per hectare. Since the authors' present study used 18 tonnes/hectare, it is on the low end of the study [59]. In order to compare the present study to [59], rather than generaly comparing pH changes, one should use the empirical equations shown in Figure 1 of [59]. Those empirical equations show two things. One is that for limestone application, the pH effect (which is the slope of the fitted lines) is in the order of 0.148 to 0.269 pH units per tonne of limestone. Correting this for the liming efficiency reported in [59] (72.5%), we get 0.204 to 0.371 pH units per tonne of CaCO3. So if we take the 0.68 tonne/hectare value stated in the present study as the rate of CaCO3 addition to the soil, we would expect to see 0.14 to 0.25 pH units increase just due to the CaCO3 amendment effect, which in in line with the observed pH changes. So study [59] helps to confirm that most of the pH effect observed is likely to be attributable to the CaCO3 content of the basalt. Notably, in study [59], they report that the effect of basalt amendment on pH was only 0.00022 pH units per tonne of basalt amendment (for one soil, for the other the effect was negative, so basalt did not prevent the soil from becoming naturally more acidic). It may be that the basalt used in [59] contained less CaCO3. This bring the second point, which is that the authors state that most basalts have no more than 1.25 wt% CaCO3, while the one used in the present study had three times this value. So more reasoning to attribute the pH effect to CaCO3 content for the duration of the present study. The manuscript already states this, deep in the discussion, but not as clearly or quantitatively as I just described. Most notably, in the abstract, it is simply stated that the pH effect observed is "likely due to rapidly dissolving minerals". This is very vague, it suggest that basalt has rapidly reactive silicates that the study found conclusively to contribute to this effect, but that is not the case. The abstract needs to be more precise on this point since it is what most readers will read.

B) EW data:

Considering that the study was done in close cooperation with an MRV company, and several authors are associated with the MRV company, it is odd that no MRV data is available to be reported in this study. As the authors state, the uptake of potassium by the plants and its accumulation in grains could be a sign of weathering, but that assumes that the element is present only in the identified silicate phases, and not as a readily dissolvable salt in the amorphous or unidentified phase. Plant availability testing (i.e., leaching tests) should be done to confirm if the release of K can really be attributed to EW or simply to the amendment or to other soil changes. Therefore, if the K uptake, is not a true sign of EW, and the soil pH change is largely due to CaCO3, is this really an EW study, or is it called an EW study because of the involvement of an MRV company?

Lets imagine that it was a different academic research group that did this experiment, without involvement of an MRV company. Either (a) they would not use the term "enhanced weathering" to describe their research (as seen in many papers that have tested basalt as a soil remineralizer or liming agent), or (b) they would have looked for more evidence of EW happening by analizing soils and porewater.

So the situation of this manuscript is overall odd in terms of how it is advancing the science, and not just how it will be a useful paper for an MRV company to refer to when saying that the technology is scientifically sound. I am especially saying this because I know that the academic group attached to this work is capable of publishing very scientifically interesting and rich studies.

Reviewer #2: (No Response)

7. PLOS authors have the option to publish the peer review history of their article (what does this mean?). If published, this will include your full peer review and any attached files.

Reviewer #1: No

Reviewer #2: No

---

## [Author Response · Author response to Decision Letter 1]

13 Nov 2023

Response to reviewer comments can be found in the file ResponseToReviewers_II.pdf

---

## [Editor Report · Decision Letter 2]

14 Nov 2023

Initial agronomic benefits of enhanced weathering using basalt: A study of spring oat in a temperate climate

PONE-D-23-15404R2

Dear Dr. Skov,

We’re pleased to inform you that your manuscript has been judged scientifically suitable for publication and will be formally accepted for publication once it meets all outstanding technical requirements.

Kind regards,

Timothy Omara, PhD

Academic Editor

PLOS ONE
---

## [Editor Report · Acceptance letter]

23 Jan 2024

PONE-D-23-15404R2 

PLOS ONE

Dear Dr. Skov, 

I'm pleased to inform you that your manuscript has been deemed suitable for publication in PLOS ONE. Congratulations! Your manuscript is now being handed over to our production team.

Kind regards, 

on behalf of

Dr. Timothy Omara 

Academic Editor

PLOS ONE